

# Gene expression profiles for *in vitro* human stem cell differentiation into osteoblasts and osteoclasts: a systematic review

Shahrul Hisham Zainal Ariffin[1], Ker Wei Lim[1], Rohaya Megat Abdul Wahab[2], Zaidah Zainal Ariffin[3], Rus Dina Rus Din[4], Muhammad Ashraf Shahidan[1], Anis Nabilah Johari[1] and Intan Zarina Zainol Abidin[5]

[1] Department of Biological Sciences and Biotechnology, Faculty of Science and Technology, Universiti Kebangsaan Malaysia, Bangi, Selangor, Malaysia
[2] Centre of Family Dental Health, Faculty of Dentistry, Universiti Kebangsaan Malaysia, Kuala Lumpur, Wilayah Persekutuan Kuala Lumpur, Malaysia
[3] School of Biology, Faculty of Applied Sciences, Universiti Teknologi MARA, Shah Alam, Selangor, Malaysia
[4] Forensic Science Programme, Faculty of Health Sciences, Universiti Kebangsaan Malaysia, Bangi, Selangor, Malaysia
[5] Centre for Research and Graduate Studies, University of Cyberjaya, Cyberjaya, Selangor, Malaysia

Corresponding authors
Shahrul Hisham Zainal Ariffin, shahroy8@gmail.com, hisham@ukm.edu.my
Intan Zarina Zainol Abidin, izzarina7@gmail.com, intanzarina@cyberjaya.edu.my

## ABSTRACT

**Background.** There have been promising results published regarding the potential of stem cells in regenerative medicine. However, the vast variety of choices of techniques and the lack of a standard approach to analyse human osteoblast and osteoclast differentiation may reduce the utility of stem cells as a tool in medical applications. Therefore, this review aims to systematically evaluate the findings based on stem cell differentiation to define a standard gene expression profile approach.

**Methods.** This review was performed following the PRISMA guidelines. A systematic search of the study was conducted by retrieving articles from the electronic databases PubMed and Web of Science to identify articles focussed on gene expression and approaches for osteoblast and osteoclast differentiation.

**Results.** Six articles were included in this review; there were original articles of *in vitro* human stem cell differentiation into osteoblasts and osteoclasts that involved gene expression profiling. Quantitative polymerase chain reaction (qPCR) was the most used technique for gene expression to detect differentiated human osteoblasts and osteoclasts. A total of 16 genes were found to be related to differentiating osteoblast and osteoclast differentiation.

**Conclusion.** Qualitative information of gene expression provided by qPCR could become a standard technique to analyse the differentiation of human stem cells into osteoblasts and osteoclasts rather than evaluating relative gene expression. *RUNX2* and *CTSK* could be applied to detect osteoblasts and osteoclasts, respectively, while *RANKL* could be applied to detect both osteoblasts and osteoclasts. This review provides future researchers with a central source of relevant information on the vast variety of gene expression approaches in analysing the differentiation of human osteoblast and osteoclast cells. In addition, these findings should enable researchers to conduct accurately and efficiently studies involving isolated human stem cell differentiation into osteoblasts and osteoclasts.

## INTRODUCTION

Bone, one of the hardest tissues in the body, serves three important functions: it provides mechanical support, acts as a shield to internal organs and carries out metabolic processes, such as providing storage for minerals and haematopoiesis (*Ansari, Ito & Hofmann , 2021*; *Konukoğlu, 2019*). As bone tissues are of paramount importance to the human body, the bones must be replenished continuously to retain their strength and structural integrity. This process is known as bone remodelling, which involves two main mechanisms, namely bone matrix formation and resorption. Osteoblasts are responsible for bone formation while osteoclasts are involved in bone resorption (*Konukoğlu, 2019*; *Phan, Xu & Zheng, 2004*). Faulty regulation of these two mechanisms disrupts the bone remodelling cycle, making them potential targets for pharmacological interventions in disease states such as osteoporosis (*Kenkre & Bassett, 2018*).

Stem cells have shown promise in tissue regeneration and have been considered for application in medicine, such as repairing defective tissues and organs, including bone tissues. Several types of mesenchymal stem cells (MSC) isolated from various organs have been suggested as a source of osteoblast progenitors, such as dental pulp tissues (*Koh et al., 2021*; *Shahrul Hisham et al., 2016*) and peripheral blood (*Shahrul Hisham et al., 2010*; *Shahrul Hisham et al., 2019*). MSC have anti-inflammatory, angiogenic and immunomodulatory properties, which are responsible for wound healing and regeneration. Preclinical and clinical studies have shown promising potential to treat degenerative diseases that involve osteoblasts and osteoclasts, including osteoporosis and osteogenesis imperfecta (*Götherström & Walther-Jallow, 2020*; *Paim & Wink, 2022*; *Yahao & Xinjia, 2021*). The use of cell-based regenerative medicine is able to modulate bone resorption, to reduce the susceptibility of fractures and to enhance the loss of mineral density (*Arjmand et al., 2020*; *Iaquinta et al., 2019*).

Osteoblasts and osteoclasts are the main cells that exist in the organic phase of bone tissue (*Iaquinta et al., 2019*). Osteoblasts differentiate from MSC that have been induced by regulatory factors such as bone morphogenic proteins (BMP). Osteoblasts produce bone matrix proteins wherein type 1 collagen (COL1A) is the most abundant extracellular protein of bone and is responsible for tissue mineralisation. Therefore, human osteoblast differentiation is observed through the expression of various kinds of bone-related extracellular matrix proteins, such as COL1A, osteocalcin (OCN), osteopontin (OPN) and bone sialoprotein (BSP). In addition, an increase in the alkaline phosphatase (ALP) activity profile is believed to be the major contributor to its characteristics (*Intan Zarina et al., 2010*; *Katagiri & Takahashi, 2002*).

Osteoclasts are formed via the fusion of monocyte lineage cells, activating bone-resorbing osteoclast cells. A myriad of factors such as cytokines, signalling molecules and transcription factors aid osteoclast differentiation. Macrophage colony-stimulating factor (M-CSF) and receptor activator of nuclear factor-$\kappa$B ligand (RANKL), which are produced by osteoblasts,

are crucial to activate osteoclast differentiation. The survival of osteoclasts is maintained through the binding of RANKL to the nuclear factor $\kappa$B receptor, which induces the formation of multinuclear osteoclasts (*Kang et al., 2014*).

Osteoblasts and osteoclasts communicate with each other through cell–cell interactions, cytokines or the cell-bone matrix. This communication occurs at various stages of differentiation, including the early and proliferative stages. Moreover, osteoblasts affect osteoclast differentiation through several pathways, such as the osteoprotegerin (OPG)/RANKL/receptor activator of nuclear factor $\kappa$B (RANK) pathway. Osteoclasts are also involved in the formation of bones by osteoblasts, where osteoclasts resorb the bone matrix (*Chen et al., 2018*).

Osteoblast and osteoclast differentiation can be observed using various gene expression profiling approaches. Gene expression profiling is the study of the gene expression pattern at the transcript level. Genes that contain biological information about the organisms are transcribed into RNA and then translated into proteins (*Brown, 2012*). Hence, the analysis of gene expression can be directly correlated to the end products of the genes. These analyses enable researchers to understand the process, development and behaviour of cells, and the interactions among cells.

There have been many publications on how gene expression approaches can help to analyse human stem cells differentiation into osteoblasts and osteoclasts. However, there are a wide variety of molecular techniques and there is currently no established standard method to analyse the differentiation of human osteoblasts and osteoclasts. Hence, this systematic review aims to collect and evaluate the findings of studies that have examined osteoblast and osteoclast differentiation. This information should help to develop a standard technique with suitable markers to investigate osteoblast and osteoclast differentiation. Moreover, it should provide researchers with a central source of information to perform more efficient and accurate experiments.

## MATERIAL AND METHODS

This systematic review was performed by following the Preferred Reporting Items for Systematic Reviews and Meta-Analyses (PRISMA) guidelines (*Page et al., 2021*). The PICOS question was established as follows: Amongst the many methodologies and genes available, which are the most suitable techniques and genes for standard analysis of human stem cell differentiation into osteoblasts and osteoclasts? Two independent observers (K.W.L. and A.N.J.) performed the searches and evaluated the articles to determine their eligibility. Three other authors (S.H.Z.A., R.M.A.W. and I.Z.Z.A.) helped resolve any discrepancies from the stated methods.

### Data search

The studies included in this systematic review were retrieved from the PubMed and Web of Science databases. Independent keywords and their combinations were applied to the search engines of these databases. A detailed customised search strategy was established for each electronic database (Table 1). The title, abstract, authors' names and affiliations, journal name and year of publication were exported to a Microsoft Excel spreadsheet. K.W.L. and

**Table 1 The combinations of the keywords used in the search.**

| Database | Search strategy |
| --- | --- |
| PubMed | {[((((molecular analysis) AND (stem cell)) AND (differentiation)) AND (osteoblast)] AND (osteoclast)} AND (human) |
| Web of Science (WOS) | ALL= (molecular analysis AND stem cell AND differentiation AND osteoblast AND osteoclast AND human) |

A.N.J. then independently screened the titles and abstracts to assess each article's eligibility for inclusion. During this phase, disagreements between the two observers were discussed and resolved by consensus. If no agreement could be reached, a third observer (S.H.Z.A.) was involved.

## Selection criteria

Only original articles that were published in the English language between 2016 and 2022 were included; review articles and duplicate articles were excluded. *In vitro* studies involving the potential of only human stem cells to differentiate into osteoblasts and osteoclasts were included; studies using any cell lines or primary cultures from animals were excluded. *In vivo* studies were not included. Studies that had combinations of both animal and human cultures were included, but only the section on human cell cultures was considered.

## Data extraction and screening process

Screening involved the following process. First, review articles and articles published in a language other than English were removed. Next, studies performed without utilising human stem cells and that did not match the parameters of osteoblastic and osteoclastic differentiation were removed. Techniques involved in screening profiles were also excluded in this review. All the remaining articles were screened for their eligibility. Data were extracted from each included article by following the PRISMA guidelines (*Page et al., 2021*). The data extracted included: study characteristics (first author, year of publication, language and study design), organism and cell lines and the methods used to analyse gene expression profiles.

## Risk of bias assessment

The quality of methodology of the included studies was evaluated by K.W.L. following the 'Modified CONSORT checklist of items for reporting *in vitro* studies', with slight modifications to fit the study. The main domains are listed as follows: (1) structured summary in abstract, (2) specific objectives or hypothesis, (3) study population, (4) further description of interventions, (5) primary and secondary outcomes, (6) results, (7) limitations, (8) sources of funding and (9) availability of protocol. Any domains that were related to clinical trials or cell lines and primary cultures from animal samples were excluded, while the domains involving human stem cells were included. Disagreements between reviewers were resolved after discussion. Each criterion was marked as follows: present (+), absent (-), unclear (?), not stated (/) or not applicable (NA).

**Table 2  Risk of bias assessment.**

| Domain<br>Author (year) | 1 | 2 | 3 | 4 | 5 | 6 | 7 | 8 | 9 |
|---|---|---|---|---|---|---|---|---|---|
| *Srikanth et al. (2016)* | + | + | + | + | + | + | / | / | + |
| *Bradamante et al. (2018)* | + | + | + | + | + | + | + | / | + |
| *Xu et al. (2018)* | + | + | + | + | + | + | + | + | + |
| *Höner et al. (2018)* | + | + | + | + | + | + | / | / | + |
| *Hashimoto et al. (2018)* | + | + | + | + | + | + | / | / | + |
| *Xie et al. (2021)* | + | + | + | + | + | + | + | + | + |

**Notes.**
[+] Domain included.
[-] Domain absent.
[/] Domain not stated in the article.
[1-9] Main CONSORT domain.

# RESULTS

## Data extraction results

Searches with keywords in the electronic databases related to stem cell osteogenic differentiation (Table 1) produced a total of 52 articles, 17 results from PubMed and 35 results from Web of Science. Every potential article was assessed independently based on the inclusion and exclusion criteria. After removing five duplicates between the two databases, 47 articles remained. A review article was also excluded. In addition, a single article in Japanese and a single article in German were excluded, followed by 12 articles not relevant to human stem cells and 26 articles that did not match the parameters of interest. Hence, there are six articles published between 2016 and 2022 eligible for qualitative synthesis. Figure 1 shows the flowchart of the article selection process.

## Study design

Of the six included studies, four focussed on the differentiation of osteoblasts, and another two focussed on the differentiation of both osteoblasts and osteoclasts. However, no studies focussed specifically on the differentiation of osteoclasts. All six studies were based on *in vitro* studies. The cultured cells included CD34+ peripheral blood stem cells (HSC) (*Srikanth et al., 2016*), and human mesenchymal stem cells (hMSC) (*Bradamante et al., 2018*; *Höner et al., 2018*; *Xie et al., 2021*; *Xu et al., 2018*). One study included multiple cultured cells, namely hMSC and human blood peripheral mononuclear cells (hBPMC) (*Höner et al., 2018*). The results from the risk of bias assessment are shown in Table 2. The included studies presented an abstract with a brief rationale and clear objective or hypotheses and introduction. The studies also provided information on the populations and results. Three studies did not state the limitations while four studies did not provide information regarding funding.

## Gene expression profiles of osteoblasts and osteoclasts

The methods used to analyse osteoblast and osteoclast differentiation included gene expression evaluated with quantitative polymerase chain reaction (qPCR) (*Hashimoto et al., 2018*; *Höner et al., 2018*; *Srikanth et al., 2016*; *Xie et al., 2021*; *Xu et al., 2018*) or microRNA

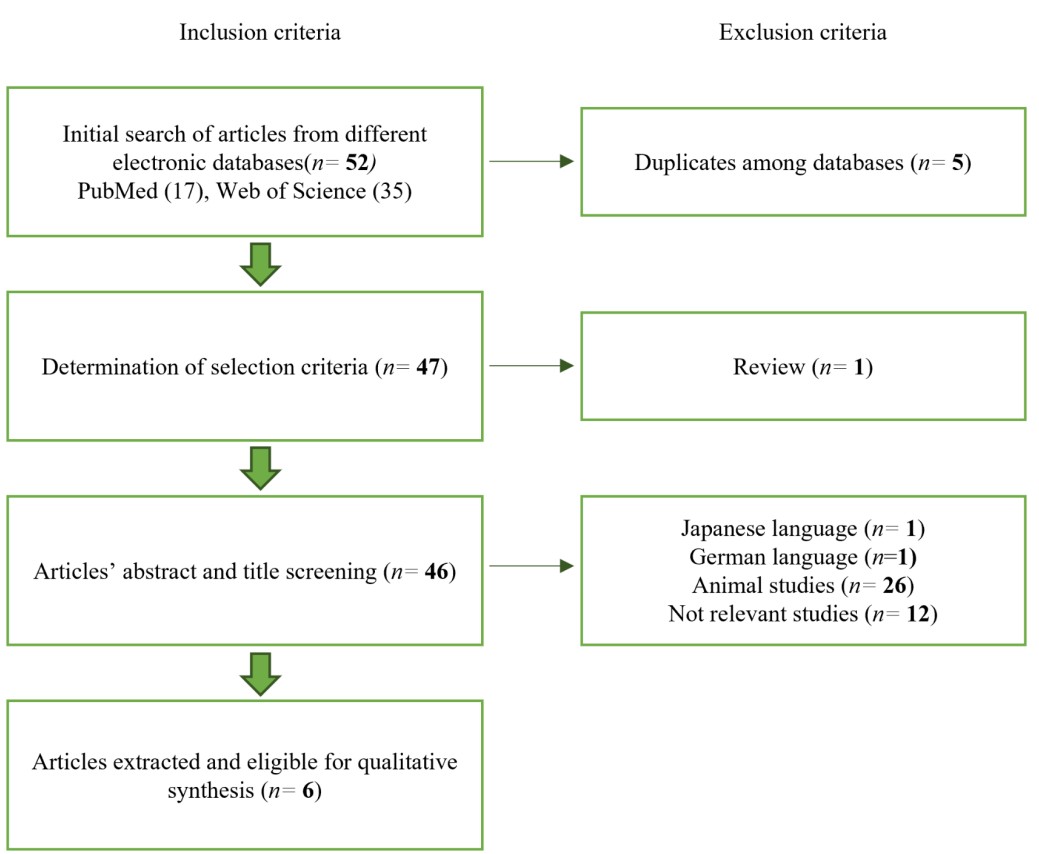

**Figure 1  Article selection process.** The article selection process was performed following the exclusion and inclusion criteria.

sequencing (miRNA-seq) analysis (*Bradamante et al., 2018*). CD34+ HSC express runt-related transcription factor 2 (*RUNX2*), osterix (*OSX*), *RANKL* and osteonectin (*SPARC*) during osteoblast differentiation. Meanwhile, when using human bone marrow-derived mesenchymal stem cells (hBMSC), *miR-142-5p* is the only gene expressed during osteoblast differentiation. hMSC, the most commonly used cells in studies included in this systematic review, express *COL1A*, *BSP*, *OPN*, *OCN*, *miR-139-5p*, *ALP*, *OPG*, *miR-940*, four and a half LIM domains 2 (*FHL2*) and *RUNX2* during osteoblast differentiation. hMSCs were also used for osteoclast differentiation; they express cathepsin K (*CTSK*), *NOTCH1*, *HES1* and *HEY1*.

Irrespective of the cell types and methodologies, there are certain gene expression profiles for osteoblast and osteoclast differentiation. These genes are specifically expressed during either osteoblast or osteoclast differentiation, except *RANKL*, which is expressed during both. *RUNX2*, *OSX*, *SPARC*, *miR-142-5p*, *COL1A*, *BSP*, *OPN*, *miR-139-5p*, *ALP*, *OPG*, miR-940 and *FHL2* are only expressed during osteoblast differentiation. *CTSK*, *NOTCH1*, *HES1* and *HEY1* are expressed specifically during osteoclast differentiation. Table 3 presents the gene expression levels, techniques and the type of stem cells applied to investigate osteoblast and osteoclast differentiation, while Table 4 includes the gene markers

used in the included studies. Based on Table 4, *RUNX2* is the most commonly expressed gene during osteoblast differentiation, while *CTSK* is the most commonly expressed gene during osteoclast differentiation.

Upregulation of *RUNX2*, *OSX*, *RANKL*, *SPARC*, *BSP*, *COL1A*, *OPN*, *OPG*, *miR-940*, *ALP*, and/or *FHL2* indicates osteoblast differentiation. *OPG* is highly expressed during osteoblast differentiation and halts osteoclast differentiation. *CTSK* is highly upregulated during osteoclast differentiation. Increased expression of Notch signalling pathway genes, including *NOTCH1*, *HEY1* and *HES1*, ultimately suppress osteoblast differentiation.

## DISCUSSION

### Types of stem cells used in osteoblast and osteoclast analysis

Of the six included studies, hMSC were the most commonly used cell type (*Bradamante et al., 2018*; *Hashimoto et al., 2018*; *Höner et al., 2018*; *Srikanth et al., 2016*; *Xu et al., 2018*). On the other hand, hMSC isolated from bone marrow, namely hBMSC (*Bradamante et al., 2018*; *Xie et al., 2021*), were the most used cell source, followed by a hMSC cell line from the human umbilical cord (*Hashimoto et al., 2018*), primary culture of hMSC from the femoral head (*Höner et al., 2018*) and primary culture of hMSC from blood peripheral monocytes (*Xu et al., 2018*). The properties of hBMSC such as ease of isolation from bone marrow without causing an immunological problem and the ability to reach confluence in a short period make them the most popular model for *in vitro* osteogenic differentiation studies (*Bhat et al., 2021*; *Ouryazdanpanah et al., 2018*). *Ansari, Ito & Hofmann (2021)* showed that rapid osteogenic differentiation under biochemical and/or mechanical stimuli significantly increase gene expression specific to osteoblast differentiation. The other variant of adult stem cells in the included studies are HSC (*Srikanth et al., 2016*). HSC are the most thoroughly characterised tissue-specific stem cells and possess potential in regenerative medicine (*Zakrzewski et al., 2019*). Monocytes derived from HSC, which comprise 10%–20% of peripheral blood, have been used during *in vitro* studies as osteoclast precursor cells. HSC and monocytes can be isolated and purified based on the expression of their specific surface markers such as CD34 and CD14. However, unlike MSC, the HSC isolation procedures are time-consuming and might lead to a low number of cells obtained, resulting in a larger volume of peripheral blood needed (*Ansari, Ito & Hofmann , 2021*).

### Gene expression profiling techniques

Genes are upregulated and downregulated during cell-specific differentiation. Changes in gene expression can be detected by qPCR and miRNA-seq analysis. qPCR can detect the expression of a single gene while miRNA-seq analysis provides a profile of predefined transcripts or genes via hybridisation. Most of the studies included in this systematic review used qPCR rather than miRNA-seq to evaluate osteoblast and osteoclast differentiation because qPCR provides quantitative information about relative gene expression. In addition, miRNA-seq lacks an optimised standard protocol despite its computational infrastructure and bioinformatic analyses (*Rao et al., 2019*). Therefore, qPCR is the most commonly chosen technique.

Zainal Ariffin et al. (2022), PeerJ, DOI 10.7717/peerj.14174

**Table 3  Gene expression profile approach.**

| Reference | Cell variant | Technique(s) used | Upregulation/ Highly expressed | | No expression | Downregulation/ Low expression | |
|---|---|---|---|---|---|---|---|
| | | | Osteoblast | Osteoclast | Osteoclast | Osteoblast | Osteoclast |
| Srikanth et al. (2016) | CD34 + peripheral blood stem cell (HSC) | qPCR | Runx2, Osterix, RANKL, SPARC | | RANK, OSCAR, NFATc, CTSK | | |
| Bradamante et al. (2018) | hBMSC | miRNA-seq analysis | miR-142-5p | | | | |
| Xu et al. (2018) | Primary culture of hMSC from blood peripheral monocyte | qPCR | BSP, COLA1, OPN, Runx2, miR-139-5p | | | | Notch1, Hey1, Hes1 |
| Höner et al. (2018) | hMSC | qPCR | Runx2, OPN, Col1, OPG, ALP | | | | |
| Hashimoto et al. (2018) | hMSC cell line from umbilical cord | qPCR | miR-940, ALP | CTSK | | | |
| Xie et al. (2021) | hMSC | qPCR | FHL2, Runx2, ALP, Col1a | | | | |

**Table 4** Summary of the gene markers for osteoblast and osteoclast cells.

| Gene markers | Frequencies used | Indication |
|---|---|---|
| Nuclear factor κb ligand (*RANKL*) | 1 | Osteoblast & osteoclast differentiation |
| Runt-related transcription factor 2 (*Runx2*) | 5 | |
| Collagen type 1 (*COL1a*) | 4 | |
| Alkaline phosphatase (*ALP*) | 2 | |
| miR-142-5p | 1 | |
| Osterix | 1 | |
| Bone sialoprotein (*BSP*) | 1 | Osteoblast differentiation |
| Osteopontin (*OPN*) | 1 | |
| miR-139-5p | 1 | |
| Osteonectin (*SPARC*) | 1 | |
| Osteoprotegerin (*OPG*) | 1 | |
| Osteocalcin (*OCN*) | 1 | |
| miR-940 | 1 | |
| Four and a half LIM domains 2 (*FHL2* gene) | 1 | |
| Cathepsin K (*CTSK*) | 2 | Osteoclast differentiation |
| Notch signalling pathway (*Notch1, Hes1, Hey1*) | 1 | |

## Markers for both osteoblast and osteoclast differentiation

*RANKL* stimulates osteoclast formation and activity, which induces the expression of *RANKL* by osteoblastic stromal cells (*Konukoğlu, 2019*; *Tobeiha et al., 2020*). RANKL together with its receptor, RANK, is essential for bone remodelling. *RANKL* is highly expressed in osteoblasts while it is also important in osteoclastogenesis: dysregulation of *RANKL* signalling may impair bone resorption (*Ono et al., 2020*). Osteoblasts regulate bone resorption through *RANKL* expression (*Konukoğlu, 2019*). RANKL, part of the RANKL/RANK/OPG signalling pathway, is secreted by osteoblasts. It then binds to its receptor (RANK) on osteoclasts and increases osteoclastic differentiation, resulting in bone resorption and bone loss (Fig. 2) (*Roumeliotis et al., 2020*). On the other hand, OPG could bind to RANKL to inhibit osteoclastogenesis (*Tobeiha et al., 2020*).

## Gene expression profile of osteoblast differentiation

The most frequently used gene markers among the included articles to detect osteoblastic differentiation are *RUNX2* (*Höner et al., 2018*; *Srikanth et al., 2016*; *Xie et al., 2021*; *Xu et al., 2018*) and *COL1A* (*Höner et al., 2018*; *Xie et al., 2021*; *Xu et al., 2018*). Early osteoblastic genes such as *RUNX2* and *OSX* showed high expression on the seventh day of culture, and enhanced expression was the key factor of osteogenesis (*Xu et al., 2018*; *Srikanth et al., 2016*). RUNX2 is a member of the Runt-related transcription factor family. It is a master transcription factor and communicates with target gene promoters via its Runt domain. RUNX2 facilitates bone remodelling through interaction with proteins and DNA sequences (*Narayanan et al., 2019*). Positive and negative regulation of *RUNX2* is crucial for bone formation (*Narayanan et al., 2019*). RUNX2 is initially detected in pre-osteoblasts and later
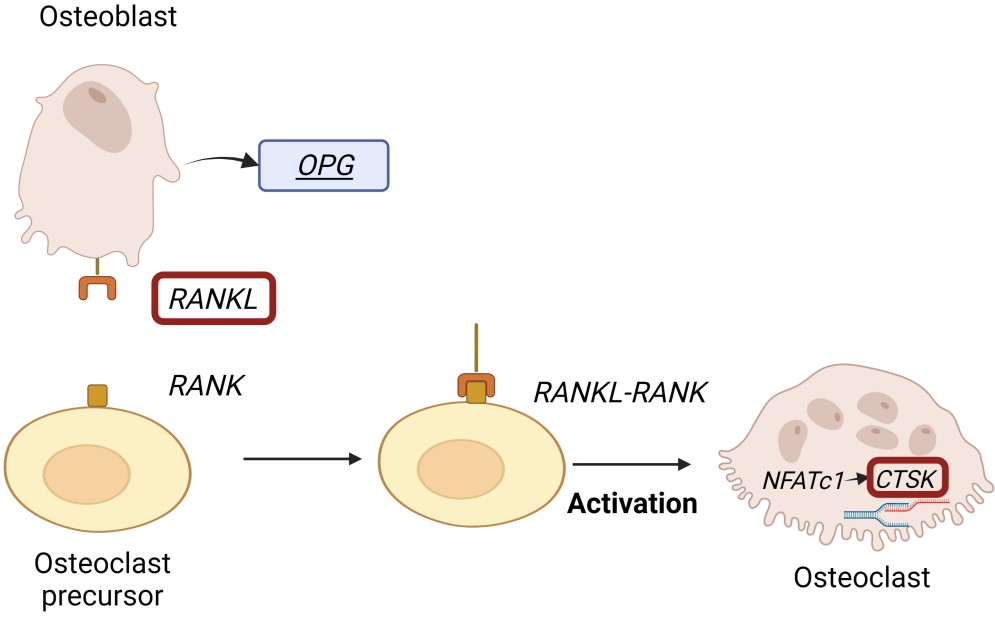

Osteoblast

Osteoclast
precursor

Osteoclast

Created with BioRender.com

**Figure 2** **RANKL/RANK interaction and CTSK expression.** Binding of RANKL, which is secreted by osteoblasts to its receptor, RANK embedded on osteoclasts results in bone resorption and bone loss. RANKL-RANK signalling pathway regulates the expression of CTSK through activation of NFATc1. Figure created with BioRender.com.

upregulated in immature osteoblasts but downregulated in mature osteoblasts (Fig. 3). RUNX2 is required for the determination of the osteoblast lineage during multipotent MSC differentiation into immature osteoblasts (*Komori, 2009*). *RUNX2* encodes multiple transcripts that are derived from two promoters (P1 and P2) and alternative splicing. P1 (distal) and P2 (proximal) initiate the expression of the major RUNX2 isoforms, type II (RUNX2-II) and type I (RUNX2-I), respectively. The structure of the promoter has been conserved in both human and murine *RUNX2* genes. RUNX2-I is expressed by osteoblasts at consistent levels throughout osteoblast differentiation while RUNX2-II expression is increased during osteoblast differentiation under the induction BMP (*Schroeder, Jensen & Westendorf, 2005*). Therefore, RUNX2 is the master transcription factor, and no bone is formed in the absence of RUNX2; making RUNX2 the preferred standard marker for osteoblast differentiation.

COL1A is a bone matrix protein that facilitates morphological changes and transformation of pre-osteoblasts into mature osteoblasts; it also serves as an early marker for osteoblasts (Fig. 3) (*Narayanan et al., 2019*). Collagen is a triple helical structure in which procollagen forms the first helical structure during collagen synthesis. Protease removes the amino and carboxyl ends of the molecule, forming tropocollagen followed by cross-linking. PYD and DYP cross-link collagen polypeptides, providing mechanical support to maintain and stabilise collagen. These cross-linkages affect the differentiation of osteoblasts. DYP is a more specific and sensitive marker as it is found specifically in

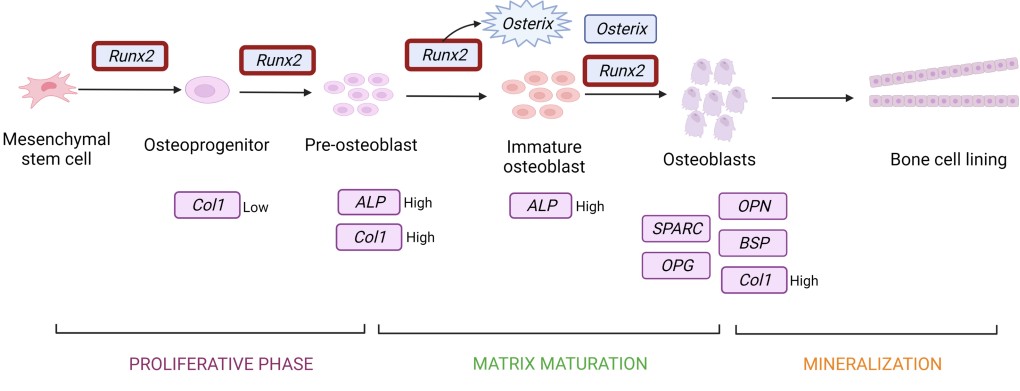

**Figure 3** **Gene expression profile of the osteoblastic cell.** *Runx2* expression is upregulated throughout from osteoprogenitor to osteoblast. However, there is a downregulation in mature osteoblast. *Runx2* activates *osterix* to produce immature osteoblast and both *Runx2* and *osterix* induce osteoblastic cells. *Col1* and *ALP* genes are expressed in osteoprogenitor to immature osteoblast under different conditions. Other genes such as *OPG, SPARC, OCN,* and *OPN* are secreted and induced mature osteoblastic cells. Figure created with BioRender.com.

bones and dentin (*Konukoğlu, 2019*). Similarly to *RUNX2*, MSC transfected with a miRNA mimic of miR-139-5p significantly enhances *COL1A* expression (*Xu et al., 2018*).

Some osteoblastic markers such as *RANKL* and *SPARC* (*Roumeliotis et al., 2020*) show high expression only at the later stages of osteoblast differentiation. SPARC regulates extracellular matrix assembly and the formation of matrix metalloproteinases and collagen that is needed for fibronectin-induced, integrin-linked kinase activation as extracellular matrix development needs an organised fibronectin matrix (*Purnachandra Nagaraju et al., 2014*).

FHL2 interacts with integrins and transcription factors to control osteoblast differentiation. *FHL2* overexpression leads to rapid differentiation of stem cells into osteoblasts and increases the expression of osteoblast markers. On the other hand, knocking out *FHL2* downregulates osteoblast markers (*Lai et al., 2006*; *Xie et al., 2021*).

*OSX* is also upregulated during osteoblast differentiation. This gene encodes an osteoblast-specific transcription factor required for osteoblast differentiation and bone formation (Fig. 3). *OSX*, one of the early osteoblastic genes, shows high expression in the early days of differentiation (*Srikanth et al., 2016*). *OSX* is considered a major effector in skeletal formation. OSX interacts with the nuclear factor of activated T cells (NFAT), which then forms a complex that improves osteoblast-mediated bone formation via activation of the *COL1A1* promoter (*Han et al., 2016*).

*ALP* (*Hashimoto et al., 2018*; *Höner et al., 2018*) is an early marker for osteoblastic differentiation produced by osteoblasts; its elevated level is positively correlated with bone formation rate (Fig. 3). ALP increases the local rate of inorganic phosphate release and aids in mineralisation while reducing extracellular pyrophosphate, which is an inhibitor of mineral formation (*Vimalraj, 2020*).

*OPN* and *BSP* are co-expressed in osteoblasts and osteoclasts. These genes encode proteins that promote adhesion of the cells to the bone matrix through the RGD (Arg-Glu-Asp) cell adhesion sequence. OPN is an acidic molecule; its central section consists of sequences that communicate and interact with seven integrins. OPN is a crucial factor in bone remodelling and settling osteoclasts in the bone matrix (*Zhao et al., 2018*). On the other hand, BSP, which is highly negatively charged, can isolate calcium ions while conserving polyglutamate regions with hydroxyapatite crystal nucleation potential. BSP allows the attachment and activation of osteoclasts through the RGD motif (*Huang et al., 2005*). qPCR analysis has revealed that elevated *OPN* and *BSP* expression is an osteogenic signature (*Xu et al., 2018*).

OPG (*Iaquinta et al., 2019*) is a decoy receptor for RANKL. It is secreted by osteoblasts to inhibit osteoclast differentiation: OPG binds to RANKL to block the interaction between RANK and RANKL (*Kenkre & Bassett, 2018*). *Kang et al. (2014)* reported very low *OPG* expression during osteoclastogenesis in osteoclasts involved in alveolar bone resorption. *OPG* expression plays a role in autoregulation in the later phase of osteoclastogenesis (*Kang et al., 2014*).

Several studies have shown the role of miRNAs in bone turnover, such as miR-940, which promotes *in vitro* osteoblast differentiation from hMSC (*Hashimoto et al., 2018*; *Konukoğlu, 2019*). However, the roles of miRNAs are very complicated, and additional studies are needed to understand them better.

## Gene expression profile of osteoclast differentiation

The Notch signalling pathway is highly conserved; it regulates cell proliferation and differentiation, determines cell fate and is involved in cellular processes in adult tissues, including skeletal tissue development and regeneration (*Luo et al., 2019*). The Notch pathway regulates bone marrow mesenchymal progenitors by suppressing osteoblast differentiation and *NOTCH1* overexpression inhibits osteoblastogenesis in stromal cells. Hence, activation of Notch signalling has a negative effect on osteoblast differentiation. When exposed to an osteogenic induction medium, MSC are forced to undergo epigenetic modifications, resulting from the upregulation of *miR-139-5p*, a phenomenon that inhibits NOTCH1 signalling activity, triggering osteoclast differentiation (*Xu et al., 2018*). However, *NOTCH1* deletion indirectly promotes osteoclast differentiation through the enhancement of osteoblast-lineage-cell-mediated stimulation of osteoclastogenesis (*Konukoğlu, 2019*).

Osteoclast Associated Ig-Like Receptor (*OSCAR*), *RANK*, *NFATC* and *CTSK* are predominantly expressed by active osteoclasts (*Konukoğlu, 2019*; *Srikanth et al., 2016*). *CTSK* expression is regulated by the RANKL/RANK signalling pathway, which is one of the important pathways for osteoclastogenesis. Activation of this signalling pathway in osteoclast precursors enhances the pro-osteoclastogenesis transcriptional factor *NFATC1*, which allows the initiation of *CTSK* transcription to occur (Fig. 2) (*Dai et al., 2020*).

## CONCLUSION

In studies published from 2016 to 2022, qPCR has been the most used technique and it is suggested as a standard approach to assess stem cell differentiation into osteoblasts and

osteoclasts because it provides qualitative information on gene expression profiles. *RANKL* has been widely used as an osteogenic marker, *CTSK* is an osteoclast marker and *RUNX2* is an osteoblast marker. This review provides useful insights on gene expression profiles for future researchers evaluating human stem cell differentiation into osteoblasts and/or osteoclasts. Identification of these gene markers should increase the efficiency of future osteogenic research, a phenomenon that should ultimately promote better therapies and medications.

## List of Abbreviations

| | |
|---|---|
| *ALP* | Alkaline phosphatase |
| **BMP** | Bone morphogenic proteins |
| **BSP** | Bone sialoprotein |
| *CTSK* | Cathepsin K |
| *COL1A* | Type 1 collagen ECM Extracellular matrix |
| *FHL2* | Four and half LIM domains 2 |
| **hBPMC** | Human blood peripheral mononuclear cells |
| **hBMSC** | Human bone marrow MSC |
| **hMSC** | Human mesenchymal stem cells |
| **HSC** | Hematopoietic stem cells |
| **M-CSF** | Macrophage colony-stimulating factor |
| **MMPs** | Matrix metalloproteinases |
| **MSC** | Mesenchymal stem cell |
| **NFAT** | Nuclear factor of activated T |
| **NGS** | Next Generation Sequencing |
| *OPG* | Osteoprotegerin |
| *OPN* | Osteopontin |
| **PRISMA** | Preferred Reporting Items for Systematic Reviews and Meta-Analyses |
| **qPCR** | Quantitative polymerase chain reaction |
| *RANKL* | Nuclear factor $\kappa$B ligand |
| **RNA** | Ribonucleic acid |
| **SPARC** | Osteonectin |

### Funding

This research was funded by the Fundamental research grant scheme (FRGS), Ministry of Higher Education, Malaysia (FRGS/1/2018/STG05/CUCMS/02/1) and (FRGS/1/2011/SG/UKM/02/13), and CUCMS Research Grant Scheme (CRGS), University of Cyberjaya (CRG/01/01/2018). The funders had no role in study design, data collection and analysis, decision to publish, or preparation of the manuscript.

## Grant Disclosures

The following grant information was disclosed by the authors:
(FRGS), Ministry of Higher Education, Malaysia: FRGS/1/2018/STG05/CUCMS/02/1.
Ministry of Higher Education, Malaysia: FRGS/1/2011/SG/UKM/02/13.
CUCMS Research Grant Scheme (CRGS).
University of Cyberjaya: CRG/01/01/2018.

## Competing Interests

The authors declare there are no competing interests.

## Author Contributions

- Shahrul Hisham Zainal Ariffin conceived and designed the experiments, performed the experiments, analyzed the data, authored or reviewed drafts of the article, and approved the final draft.
- Ker Wei Lim conceived and designed the experiments, performed the experiments, analyzed the data, prepared figures and/or tables, authored or reviewed drafts of the article, and approved the final draft.
- Rohaya Megat Abdul Wahab conceived and designed the experiments, performed the experiments, analyzed the data, authored or reviewed drafts of the article, and approved the final draft.
- Zaidah Zainal Ariffin conceived and designed the experiments, performed the experiments, prepared figures and/or tables, and approved the final draft.
- Rus Dina Rus Din conceived and designed the experiments, prepared figures and/or tables, and approved the final draft.
- Muhammad Ashraf Shahidan conceived and designed the experiments, performed the experiments, prepared figures and/or tables, and approved the final draft.
- Anis Nabilah Johari conceived and designed the experiments, performed the experiments, prepared figures and/or tables, and approved the final draft.
- Intan Zarina Zainol Abidin conceived and designed the experiments, authored or reviewed drafts of the article, and approved the final draft.

## Data Availability

The PRISMA checklist is available in the Supplementary File.

## Supplemental Information

Supplemental information for this article can be found online at http://dx.doi.org/10.7717/peerj.14174#supplemental-information.

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
