# Peer review of "Gene expression profiles for in vitro human stem cell differentiation into osteoblasts and osteoclasts: a systematic review"

_PeerJ, doi:10.7717/peerj.14174_

## Round 0.1 · original submission · Major Revisions

The reviewers have given their views on the manuscript and I agree with the decision that it requires major revisions. The experimental design and techniques used require more careful revisions and the title and conclusion should reflect the information given in the methods.

Reviewer 1 ·

Basic reporting

1. This is my main concern of the manuscript. Normally, three gene markers are used to identify/confirm a biological process. Hence, I believe the authors could re-analyze the data and justify why certain genes are more suitable for use based on the findings in terms of biological processes.

2. The second issue is the use of the term DNA microarray analysis in discussion section lines 254-257. The correct type of microarray that profile gene expression is 'microarray gene expression.'

3. The manuscript's professional English could be improved. Lines 287–296 of Runx's description are examples of where English could be improved. Another example is the final sentence of lines 241 and 242, which can be rewritten to make it more clear.

4. The introduction section requires more details, particularly in explaining the context of the manuscript in order to reflect the title. I believe that the current description of stem cells used in osteogenesis or regenerative medicine should be improved. Please also elaborate on the relationship between osteoblast and osteoclast (there is a knowledge gap).

Experimental design

1. The scope of the research is suitable for this journal.

2. My concern on this part is the used of 'molecular analysis' as the keyword for the study. Therefore, in the results part there are microarray and NGS. As we all understand, these two techniques are more towards screening profiles rather than identification of gene expression profile. Therefore, I would suggest that the experimental design presentation of this manuscript be altered or to remove those two techniques from the analysis.

3. Please explain the slight modification in the assessment in the quality of methodology used in the included studies for risk of bias assessment.

Validity of the findings

1. Impact and novelty of the study is not assessed.
2. All the data have been provided, but types of stem cells were not described in the results section.
3. However, I believe the authors could rewrite some of the material to make it clearer. As examples, the abstract's results section indicated only two genes, one for osteoblast and one for osteoclast gene markers. However, four genes are named in the conclusion section.

Additional comments

Instead of describing in detail the role of each of the genes, the discussion section should be rewritten to support the findings/results, examples lines 281-300.

Reviewer 2 ·

Basic reporting

The introduction is well written with enough background information and appropriate references. The authors are described the role and formation of osteoblasts and osteoclasts which is very useful for the lay audience. The tables are informative and is east to follow; however, the quality of the diagrams could be improved. The font size is small and is hard to read if not zoomed in. The references are suitably cited, and the authors have used a good balance of old and new references.

Experimental design

This review falls in the scope of this journal. The methods of the literature review is described systematically and appears to be replicable. The authors have clearly stated the question in hand for the review and have used appropriate methods to answer the question.

Validity of the findings

The manuscript includes a healthy discussion and has answered the question that qPCR is the most used technique for gene expression to detect differentiation of osteoblast and osteoclast cells and that qPCR could become a standardize technique for it. This review also identifies genes that can be used for detecting osteoblast and osteoclast cells. This is a novel and impactful work.

Additional comments

Except for so minor changes in figures, this is a well written, systematic and important review which may provide a useful insight for future researchers studying osteoblast and osteoclast cells and their gene expression.

---

## Round 0.2 · accepted · Accept

This paper is accepted for publication.

Reviewer 1 ·

Basic reporting

The write-up has been improved.

Experimental design

The description of experimental design has been improved.

Validity of the findings

No comment.

Additional comments

No comment.

Reviewer 2 ·

Basic reporting

The authors have incorporated all the changes that were asked for.

Experimental design

NA

Validity of the findings

NA

Additional comments

NA